# Detection of *Anaplasma phagocytophilum* in Wild and Farmed Cervids in Poland

**DOI:** 10.3390/pathogens10091190

**Published:** 2021-09-14

**Authors:** Anna W. Myczka, Żaneta Steiner-Bogdaszewska, Katarzyna Filip-Hutsch, Grzegorz Oloś, Michał Czopowicz, Zdzisław Laskowski

**Affiliations:** 1Witold Stefański Institute of Parasitology, Polish Academy of Sciences, Twarda 51/55, 00-818 Warsaw, Poland; zaneta.steiner@gmail.com (Ż.S.-B.); katarzyna.filip@twarda.pan.pl (K.F.-H.); laskowz@gmail.com (Z.L.); 2Institute of Environmental Engineering and Biotechnology, University of Opole, Kardynała B. Kominka 6, 6a, 45-032 Opole, Poland; golos@uni.opole.pl; 3Division of Veterinary Epidemiology and Economics, Institute of Veterinary Medicine, Warsaw University of Life Sciences-SGGW, Nowoursynowska 159c, 02-776 Warsaw, Poland; mczopowicz@gmail.com

**Keywords:** *Anaplasma phagocytophilum*, *16S* rDNA, wild cervids, farm animals

## Abstract

Background: The role of cervids in the circulation of *A. phagocytophilum* has not yet been clearly determined; however, several species of wild and farm cervids may be a natural reservoir of this bacteria. Methods: Spleen and liver tissue samples were taken from 207 wild (red deer, roe deer, fallow deer and moose) and farmed cervids (red deer and fallow deer) from five geographical areas. These were tested for the *A. phagocytophilum*
*16S* rDNA partial gene by nested PCR. Results: *Anaplasma* spp. were detected in 91 of 207 examined cervids (prevalence 43.9%). Three different variants of *16S* rDNA partial gene were reported, one for the first time. *Anaplasma phagocytophilum* was more often detected in young specimens than in adults and more often in the spleen than in the liver. Conclusions: Cervids from the four sites across Poland were found to be major natural reservoirs of various strains of *A. phagocytophilum*. This is the first study to use spleen and liver as biological material to detect *A. phagocytophilum* in moose in Poland.

## 1. Introduction

*Anaplasma phagocytophilum* is a Gram-negative, obligate intracellular bacterium that lives mainly in neutrophils. It is a causative agent of HGA (Human Granulocytic Anaplasmosis), TBF (Tickborne Fever) in domestic animals and livestock, and GA (Granulocytic Anaplasmosis) in wild animals [1]. In animals, it was first discovered in sheep (*Ovis aries*) in Scotland, as *Cytoecetes phagocytophila* [2], while it was first reported in humans as *Ehrlichia* spp. in the USA in 1994 [3]. In wild and domestic ruminants, infection with *Anaplasma phagocytophilum* is asymptomatic and does not significantly affect the condition of the animals. However, in severe disease, the symptoms usually include weight loss, reduced milk yield, anorexia and apathy [4,5], with few fatal cases reported in sheep, roe deer (*Capreolus capreolus*) and moose (*Alces alces*) [6,7]. In humans, the symptoms are headache, dizziness, fever, abdominal pain and diarrhea [8]. In both humans and animals, the symptoms are not very specific, which makes it difficult to diagnose anaplasmosis correctly. Since the first cases over 90 years ago, *A. phagocytophilum* has been the subject of interest in both human and veterinary medicine.

The bacteria *Anaplasma phagocytophilum* is transmitted by several tick species, including those of *Ixodes* (*I. ricinus, I. pacificus, I. scapularis, I. persulcatus*) [9,10] and *Dermacentor* spp. (*D. reticulatus, D. albipictus*) [11,12]. An infected tick can transmit *A. phagocytophilum* to a range of vertebrate hosts including small mammals, wild cervids and carnivores, wild boars, horses, cattle and domestic animals [1,13,14,15,16,17,18,19]. The role of wildlife in the circulation of *A. phagocytophilum* is yet to be clearly determined, but several species of wild ruminants are thought to be important reservoirs [12].

The population of red deer (*Cervus elaphus*), roe deer (*Capreolus capreolus*), fallow deer (*Dama dama*) and other game animals continues to increase in Poland and Europe [20,21,22,23]. Moose (*Alces alces*), like other large wild cervids, had also been classified as game animals until 2001, when the government imposed a ban on hunting; since then the moose population has recovered and the population is currently estimated to be approximately 28,000 individuals [24]. The past few years have also seen a rise in the number of deer farms in response to increased demand for venison [25], and according to the Federation of European Deer Farmers Associations, at least 200 deer farms are believed to be in operation in Poland [FEDFA accessed on 08 March 2021]. In Europe, the most common species of farmed deer are red deer and fallow deer [26]. While such farming is still a relatively new trend in Poland, the country is a leading producer of farmed venison in Europe [27]. As such, the growing ecological and economic aspects of forest and livestock management have increased the need for molecular research of pathogens [28].

Detection and genotyping of *Anaplasma phagocytophilum* can lead to determining which strains of this bacterium may or may not be pathogenic for animals and humans and additionally how *A. phagocytophilum* circulates and spreads in the environment [29]. The most common genetic markers used for genotyping *A. phagocytophilum* are: *msp4*, *16S* rDNA, *groEL* and *ankA* [30]. In this study, to detect *Anaplasma phagocytophilum* in samples from wild and farmed cervids, *16S* rDNA marker was used. The aim of this study was detection of *Anaplasma phagocytophilum* in farmed and wild populations of cervids in Poland. It examines the significance of sex and age on the occurrence of *A. phagocytophilum* infection, and determines whether the spleen or liver is a more appropriate tissue for detection.

## 2. Results

### 2.1. Prevalence of Anaplasma spp.

Out of 207 examined cervids (165 wild, 42 farmed), evidence of infection with *Anaplasma* spp. was detected in 91 individuals, with a prevalence of 43.9% (44.0%, CI 95%: 37.4%, 50.8%). No positive samples were detected in fallow deer (0/36, 6 wild, 27 farmed). Among the 90 tested red deer (75 wild, 15 farmed), genetic material of *Anaplasma* spp. was detected in 50 individuals: 47 wild animals and three farmed animals. The total prevalence among all red deer was therefore 55.6% (CI 95%: 44.1%, 66.8%): 62.8% (CI 95%: 51.4%, 72.7%) for wild ones and 20% (CI 95%: 7.0%, 45.2%;) for farmed red deer. In addition, *Anaplasma* spp. DNA was detected in 39 of the 70 examined roe deer (39/70, 55.7%) and in two of 11 moose (2/11, 18.2%) (Table 1 and Table 2).

The statistical analysis did not show any significant difference in the prevalence of *A. phagocytophilum* between red deer and roe deer (*p* = 0.989). Prevalence was significantly lower in fallow deer than in red deer (*p* < 0.001), roe deer (*p* < 0.001) and moose (*p* = 0.001). Moreover, prevalence in red deer and roe deer were significantly higher than in moose (*p* = 0.019 and 0.021, respectively) (Table 2). Prevalence did not differ significantly between young and adult individuals of any species (*p* = 0.104 for red deer, *p* = 0.904 for roe deer, *p* > 0.999 for fallow deer, and *p* = 0.182 for moose), however it was significantly higher in young individuals when all species were analyzed together (*p* = 0.030) (Table 1). No statistically significant differences were found between male and female adults, neither when analyzed separately for each species (*p* = 0.470 for red deer, *p* = 0.117 for roe deer, *p* > 0.999 for fallow deer, and *p* > 0.999 for moose) nor for all adult animals together (*p* = 0.069).

### 2.2. Agreement between Spleen and Liver Samples

Of 172 animals in which both spleen and liver samples were tested, 81 (47.1%) of the animals tested positive in at least one material; however, only 23 animals (28.4% of 81 positive) were positive in both materials. *A. phagocytophilum* was significantly more often isolated from the spleen (66/172; 38.4%) than from the liver (38/172; 22.1%; *p* < 0.001). The agreement beyond chance between results obtained on the spleen and liver samples was moderate (Gwet’s AC_1_: 41.7%; CI 95%: 29.5%, 53.9%).

### 2.3. 16S rDNA Anaplasma phagocytophilum

Sequences of *A. phagocytophilum*
*16S* rDNA were obtained from three species: red deer, roe deer and moose. Nine of the obtained sequences submitted to GenBank were identical to each other and were obtained from all three species (*Cervus elaphus*: MZ314415, MZ317900, MZ317903, MZ317901; *Capreolus capreolus*: MZ317898, MZ317904, MZ317899; *Alces alces*: MZ317902, MZ317897). These sequences were found to have 100% similarity to many *Anaplasma phagocytophilum* submissions across: Europe, Asia, North America and Africa from various hosts: tick, domestic animals, farm animals, wild animals (ungulates and carnivores), rodents and humans (Table 3). Three sequences obtained from roe deer (MZ314417, MZ319389, MZ319390) were identical (100% similarity) to only one sequence of *A. phagocytophilum* from roe deer in Spain (MN170723.1) (Appendix A). Altogether these four sequences from roe deer may belong to the newly described *A. phagocytophilum* Roe deer strain (reference sequence MZ314417). Moreover, from red deer, one sequence of *A. phagocytophilum,* was found without 100% similarity to any submission in GenBank (high score Percent Identity was 99.75% to *A. phagocytophilum* MN170723.1) and presented a two nucleotide difference of *A. phagocytophilum* reference sequence (NR_044762.1) (Appendix A). The sequence was submitted to GenBank with accession number MZ314416. All sequences submitted to GenBank are included in Table 4. Variable sites in *16S* rDNA partial gene sequences can be seen in Appendix A.

## 3. Discussion

Studies performed across Europe on the potential of free-living and farmed cervids as natural reservoirs of *Anaplasma phagocytophilum* have shown these reservoirs to vary depending on the geographical region [5,18,28]. In this study, positive samples for *Anaplasma* spp. were detected in at least one of the analyzed species from four geographical areas in Poland. Only in individuals from Polesie National Park (Lublin Voivodeship) non positive samples were found.

The infection rate of *A. phagocytophilum* in all cervids was shown to be lower than in other European countries [4,18,28,30,31,32,33], lower rates, compared to the results of this study were recorded only in Spain, Czech Republic and Italy [31,34,35,36]. Outside Europe, *A. phagocytophilum* has been detected in deer from Japan, with a prevalence ranging from 15.6 to 75.4%; our results lie in the middle of this range [37,38]. However, in Poland, the prevalence is usually lower [39,40,41,42] or at a similar level [43,44].

Genetic marker *16S* rDNA is most commonly used to detect *Anaplasma phagocytophilum* among wild and farm animals [17,18,36,39,44]. In this study, three different variants of partial *16S* rDNA genetic marker were found (Appendix A). One genotype from this group (MZ317901) is very common in the natural environment and on the analyzed fragment is identical with the reference sequence of *16S* rDNA *A. phagocytophilum* (NR_044762.1) and with many other *A. phagocytophilum* sequences from various hosts and geographical regions (Table 3). The second variant of described sequences, detected from three roe deer (MZ314417, MZ319389, MZ319390) has one nucleotide change according to reference sequence of *16S* rDNA *A. phagocytophilum* (Appendix A). Additionally, these three sequences of partial *16S* rDNA genetic marker were 100% identical to the sequence from roe deer in Spain (MN170723) [45]. According to our results (Table 4) and the results of Remesar et al. (2020), this strain of *A. phagocytophilum* was detected in three different geographical regions from one animal species—roe deer. This may lead to the conclusion that this strain of *A. phagocytophilum* can be characteristic for bacteria isolated from roe deer. However, to confirm this fact more roe deer samples should be examined.

In the present study, the prevalence of *A. phagocytophilum* in farmed red deer was 20%. Although this rate is significantly lower than that previously reported by Hapunik et al. (2011) [44] in Poland, it is similar to the rate observed in farmed red deer from Germany [18]. In fallow deer, the lack of detection of *A. phagocytophilum* may be due to the fact that the majority (27/36, 75%) of the tested samples came from farmed animals, where the prevalence was typically relatively low [40,44]. In China, a similar study about the prevalence of *Anaplasma* spp. in wild and farmed cervids reported a lack of positive samples of *A. phagocytophilum* in 68 tested farmed deer [46]. Such a low prevalence, or lack of detection, of *A. phagocytophilum* in farmed animals may be due to the fact, that unlike wild animals, these animals have regular and constant access to feed and are regularly dewormed (twice a year to ectoparasites and endoparasites), which strengthens their condition and makes them less susceptible to infection by *Anaplasma phagocytophilum*. They are much less exposed to tick attacks than wild animals, by regular mowing pastures, minimal contact with wild forest animals (mainly the presence of small and medium-sized rodents is noted), lack of forest coverage and low density of farmed animals (about 8–10 individuals/10,000 m^2^). The lower number of vectors in the environment may also reduce the prevalence of *A. phagocytophilum* among these animals [44].

Only one previous report has examined the prevalence of *A. phagocytophilum* in moose in Poland. Karbowiak et al. (2015) [41] reported one infected individual (1/7, 14.3%) based on blood samples. In comparison, the present study examined samples that were from the spleen and liver. Our findings indicated that *A. phagocytophilum* was present in both tissues (2/11, 18.18%), and this is the first confirmation of the presence of *A. phagocytophilum* in peripheral tissues in moose in Poland. Comparing results from this study with results from Karbowiak et al. (2015) [41], all types of samples (spleen, liver, blood) are suitable for detection of *A. phagocytophilum*.

*Anaplasma phagocytophilum* was more than twice as likely to be isolated and detected in the spleen compared to the liver. However, from 91 individuals that tested positive, 19 samples were only from liver samples (without positive result from spleen or lack of spleen sample) and 19 out of 91 positive samples regarded for more than 20% of all positive samples. Therefore, when carrying out this type of analysis with the use of peripheral tissue, based on the obtained results, it is suggested, if possible, to use a larger number and variety of samples than only one from one individual. In Poland the most frequently chosen material for this type of research was blood [39,40,41,44,47], although sometimes spleen tissue was used to detect *A. phagocytophilum* in Poland [42]. However, recent reports by Hornok et al. (2018) [28] and Kazimirova et al. (2018) [48] showed that more positive samples with *Anaplasma phagocytophilum* detected in wild cervids species came from spleen samples than from whole blood samples.

In the present study, wild red deer and roe deer more frequently tested positive for *Anaplasma phagocytophilum* than the other species, viz wild fallow deer, moose, farmed red deer and farmed fallow deer. In contrast to previous studies, where no differences in *Anaplasma phagocytophilum* infection rate were reported between male and female cervids [18,33,49], our present findings identified a difference at the borderline of statistical significance, suggesting that *A. phagocytophilum* may be more common in females. In addition, a higher prevalence was noted in young individuals, which is in line with reports from Germany [18,33,49]. The regularity of the higher incidence of *Anaplasma phagocytophilum* among young individuals was observed in the analysis of all samples (all cervids) examined in this study. Higher prevalence of *A. phagocytophilum* among young individuals than in adults may be due to the fact that young cervids can have an underdeveloped immune system, which may facilitate the development and maintenance of *Anaplasma phagocytophilum* in their organisms. Additionally, a similar correlation between the higher prevalence of parasites among juveniles was observed in cervids with nematodes from the Protostrongylidae family [50].

## 4. Materials and Methods

### 4.1. Materials

Spleen and liver samples from free-ranging red deer, roe deer, moose and fallow deer (n = 207, 165 wild, 42 farmed) were collected in years 2017–2020 in five geographical areas: Pisz Forest (Warmian-Masurian Voivodeship), Bolimów Forest (Łódź Voivodeship), Kampinos National Park (Masovian Voivodeship), Warsaw Urban Forest (Masovian Voivodeship), Polesie National Park (Lublin Voivodeship) and Stobrawa-Turawa Forest (Opolskie Voivodeship) (Figure 1). Samples from wild game animals were collected during hunting season. Samples from moose (protected animal), were secured on an ongoing basis when there was an opportunity to collect this material and were collected from road kill animals or those found dead by forest and national park employees. In addition, spleen and liver materials were collected from farmed red deer and fallow deer from the Research Station of the Institute of Parasitology, Polish Academy of Sciences in Kosewo Górne (Warmian-Masurian Voivodeship) (Appendix A). Demographic characteristics of the study population are presented in Table 5.

### 4.2. Molecular Methods

DNA from spleen and liver was isolated using a commercial DNA Mini Kit (Syngen, Poland) according to the manufacturer’s protocol. *Anaplasma* spp. was detected using semi-nested PCR to amplify the partial *16S* rDNA gene with specific primers: for the primary reaction: A500 F 5′CGTTGTTCGGAATTATTGGGCGTA-3′, A900 R 5′-CCATGCAGCACCTGTGCGAG-3′ and for the semi-nested reaction: A520 F 5′-GGGCATGTAGGCGGTTCGGT-3′, A900 R 5′-CCATGCAGCACCTGTGCGAG-3′ [17]. DNA isolated from wild boar (*Sus scrofa*) infected with *A. phagocytophilum* (MT510541.1) was used as a positive control [19]. Nuclease-free water was added to the PCR mix as a negative control. The PCR products were visualized on a 1.2% agarose gel (Promega, Madison, WI, USA) stained with SimplySafe (EURx, Gdańsk City, Poland) and a size-marked DNA Marker 100 bp LOAD DNA ladder (Syngen, Wrocław, Poland). Visualization was performed using ChemiDoc, MP Lab software (Imagine, BioRad, Hercules, CA, USA). The obtained PCR products were purified with the DNA clean-up Kit (Syngen, Wrocław, Poland). The purified products were sequenced by Genomed (Warsaw, Poland) and assembled using ContigExpress, Vector NTI Advance v.11.0 (Invitrogen Life Technologies, New York, NY, USA). The obtained sequences were compared with sequences from GenBank in BLAST (NCBI, Bethesda, MD, USA) and submitted to GenBank.

### 4.3. Statistical Analysis

Categorical variables were presented as counts and percentages. Percentages were compared between groups using the maximum likelihood G test or Fisher’s exact test (if the expected count in any cell of the contingency table <5) for unpaired groups and McNemar’s test for paired groups. The 95% confidence intervals (CI 95%) for percentages were calculated using the Wilson score method [51]. The agreement beyond chance between results obtained on different biological materials was assessed using Gwet’s AC_1_ coefficient [52] and classed according to Landis and Koch (1977) [53]. A significance level (α) was set at 0.05 and all statistical tests were two-tailed. The statistical analysis was performed in TIBCO Statistica 13.3 (TIBCO Software Inc., Palo Alto, CA, USA).

## 5. Conclusions

*Anaplasma phagocytophilum* infection seems to be common in red deer and roe deer in Poland. It occurs more often in free-living than in farm red deer. *A. phagocytophilum* is more likely to be detected in spleen than in liver samples, and results obtained on these two types of samples are only moderately consistent. The prevalence of infection seems to be lower than in previous studies, but it is still high in the natural environment. To our knowledge, this is the first study using spleen and liver as biological material to detect *A. phagocytophilum* in moose in Poland.

## Figures and Tables

**Figure 1 pathogens-10-01190-f001:**
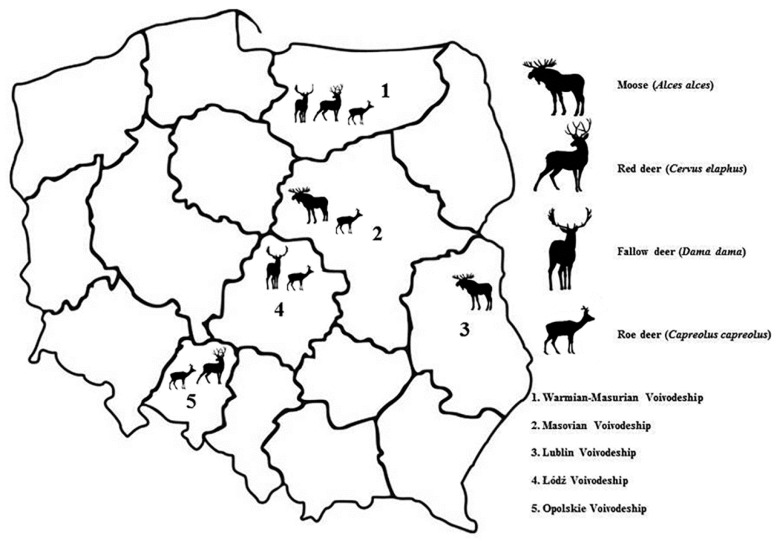
Geographical distribution of the collected samples in Poland.

**Table 1 pathogens-10-01190-t001:** The prevalence of *Anaplasma phagocytophilum* in age classes.

Species	Young ^1^		Adults		*p*-Value
	No. of Positive Animals/No. of All Young Animals	Prevalence (CI 95%)	No. of Positive Animals/No. of All Adult Animals	Prevalence (CI 95%)	
Red deer (n = 90)	22/33	66.7 (49.6, 80.2)	28/57	49.1 (36.6, 61.7)	0.104
Roe deer (n = 70)	8/14	57.1 (32.6, 78.6)	31/56	55.4 (42.4, 67.6)	0.904
Fallow deer (n = 36)	0/5	0 (0, 43.4)	0/31	0 (0, 11.0)	>0.999
Moose (n = 11)	2/5	40.0 (11.8, 76.9)	0/6	0 (0, 39.0)	0.182
Overall (n = 207)	32/57	56.1 (43.3, 68.2)	59/150	39.3 (31.9, 47.3)	0.030 *

* Difference significant at α = 0.05. ^1^ Age of animals judged by hunters and based on evidence of animals in Research Station in Kosewo Górne.

**Table 2 pathogens-10-01190-t002:** The prevalence of *Anaplasma phagocytophilum* in adult animals.

Species	Females		Males		*p*-Value
	No. of Positive Animals/No. of Females	Prevalence (CI 95%)	No. of Positive Animals/No. of Males	Prevalence (CI 95%)	
Red deer (n = 57)	23/49	46.9 (33.7, 60.6)	5/8	62.5 (30.6, 86.3)	0.470
Roe deer (n = 56)	25/49	51.0 (37.5, 64.4)	6/7	85.7 (48.7, 97.4)	0.117
Fallow deer (n = 31)	0/9	0 (0, 29.9)	0/22	0 (0, 14.9)	>0.999
Moose (n = 6)	0/3	0 (0, 56.2)	0/3	0 (0, 56.2)	>0.999
Overall (n = 150)	48/110	43.6 (34.7, 53.0)	11/40	27.5 (16.1, 42.8)	0.069

**Table 3 pathogens-10-01190-t003:** Examples of *A. phagocytophilum* sequences with 100% similarity from the NCBI GenBank.

Continent	Host		Country	GenBank No.
EUROPE	Red fox (*Vulpes vulpes*)		Switzerland	KX180948.1
			Poland	MH328211.1
	Dog (*Canis lupus familiaris*)		Croatia	KY114936.1
			Germany	JX173651.1
	Sheep (*Ovis aries*)		Norway	CP015376.1
	Red deer (*Cervus elaphus*)		Slovenia	KM215243.1
	Roe deer (*Capreolus capreolus*)		Spain	MN170723.1
	Tick (*Ixodes ricinus*)		Estonia	MW922755.1
			Belarus	HQ629915.1
			Austria	JX173652.1
	European badger (*Meles meles*)		Poland	MH328211
	Wild boar (*Sus scrofa*)		Poland	MT510541.1
	Bank vole (*Clethrionomys glareolus*)		United Kingdom	AY082656.1
	European hedgehog (*Erinaceus europaeus*)		Germany	FN390878.1
	Human (*Homo sapiens*)		Austria	KT454992.1
			Belgium	KM259921.1
AFRICA	Dog (*Canis lupus familiaris*)		Republic of South Africa (RSA)	MK814406.1
	Natal multimammate mouse (*Mastomys natalensis*)		RSA	MK814411.1
NORTH AMERICA	Tick (*Ixodes pacificus*)		United States of America (USA)	KP276588.1
	Llama (*Lama glama*)		USA	AF241532.1
	Horse (*Equus ferus caballus*)		USA	AF172166.1
	Coyote (*Canis latrans*)		USA	AF170728.1
	Human (*Homo sapiens*)		USA	AF093788.1
ASIA	Tick	*Ixodes persulcatus*	Russia (Irkutsk region)	HM366584.1
		*Ixodes ricinus*	Turkey	FJ172530.1
		*Haemaphysalis longicornis*	China	KF569908
			South Korea	GU064898
	Northern red-backed vole (*Myodes rutilus*)		Russia (Sverdlovsk region)	HQ630622.1
	Dog (*Canis lupus familiaris*)		Iraq	MN453475.1
			Japan	LC334014.1
	Raccoon dog (*Nyctereutes procyonoides*)		South Korea	KY458557.1
	Black-striped field mouse (*Apodemus agrarius*)		South Korea	KR611719.1
			China	GQ412337
				DQ342324
	Cat (*Felis catus*)		South Korea	KR021165.1
	Cow (*Bos taurus taurus*)		Turkey	KP745629.1
	Goat (*Capra hircus*)		China	KF569909.1
	Human (*Homo sapiens*)		South Korea	KP306520.1

**Table 4 pathogens-10-01190-t004:** Accessions numbers and descriptions of nucleotide sequences submitted to GenBank from this study.

GenBank No.	Host	Sample	Isolation Source	GenBank No.	Region of Origin
MZ314415	Red deer	J23	Spleen	MZ314415	Pisz Forest
MZ317900	Red deer	J28	Spleen	MZ317900	Pisz Forest
MZ317903	Red deer	J75	Liver	MZ317903	Pisz Forest
MZ317901	Red deer	J80	Liver	MZ317901	Pisz Forest
MZ314416	Red deer	J70	Spleen	MZ314416	Pisz Forest
MZ317902	Moose	L9	Liver	MZ317902	Warsaw Urban Forest
MZ317897	Moose	L6	Spleen	MZ317897	Warsaw Urban Forest
MZ314417	Roe deer	S63	Spleen	MZ314417	Strobawa–Turawa Forest
MZ317898	Roe deer	S36	Spleen	MZ317898	Bolimów Forest
MZ317904	Roe deer	S24	Liver	MZ317904	Pisz Forest
MZ317899	Roe deer	S42	Spleen	MZ317899	Pisz Forest
MZ319389	Roe deer	S7	Liver	MZ319389	Pisz Forest
MZ319390	Roe deer	S8	Spleen	MZ319390	Pisz Forest

**Table 5 pathogens-10-01190-t005:** Demographic characteristics of the study population. The numbers of farmed cervids presented in parentheses.

Species	Adults ^1^		Young ^1^	Total
	Males	Females		
Red deer	8 (0)	49 (15)	33 (0)	90 (15)
Fallow deer	22 (20)	9 (2)	5 (5)	36 (27)
Roe deer	7	49	14	70
Moose	3	3	5	11

^1^ Age of animals judged by hunters and based on evidence of animals in Research Station in Kosewo Górne.

## Data Availability

The data that support the findings of this study are available from the corresponding author, upon reasonable request.

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
