# Peer review of "Detection of Anaplasma phagocytophilum in Wild and Farmed Cervids in Poland"

_pathogens, 2021, doi:10.3390/pathogens10091190_

Round 1

Reviewer 1 Report

The article is very interesting and needful. However in general, the title and aims of the study do not match the content of the article as it shlould be - see my comments in the article. Please try to improve it.

Reviewer 2 Report

In this paper, the authors clearly showed that ungulates from the five sites across Poland were the major natural reservoirs of various strains of Anaplasma phagocytophilum. This paper is well-written; however, in order for the readers to assess correctly the usefulness of spleen and liver as biological material to detect A. phagocytophilum in cervids, several points described below may be better to be re-considered.

Major point:

  1. The authors mention in the introduction section that the population of cervids is on the rise. Are there any effects on humans due to this, such as an increase in the number of people affected by HGA in Poland, or transmission of A. phagocytophilum from cervids to humans without tick bites?

  1. Lines 85-86: The authors showed that it (the prevalence of A. phagocytophilum among cervids) was significantly higher in young individuals when all species were analyzed together. Kindly consider adding an explanation of why the prevalence was higher in the young.

  1. Lines 151-152: It is written here that all type of samples (spleen, liver, blood) are suitable for detection phagocytophilum. If stored blood samples are available, it is desirable to show the data obtained from the blood sample in addition to the data from the spleen and liver.

  1. What are the benefits of using the spleen or liver as a PCR sample to detect A. phagocytophilum? Is there any advantage to using the spleen and liver samples for testing PCR instead of blood? Kindly consider adding a discussion of suitability of spleen and liver as PCR specimens compared to blood.

Minor point

  1. Line 171: Do Table A1 and Table S1 mean the same one?

  1. Informed Consent Statement (Line 219): Please correct the description in this section.

  1. Please correct any errors in the references, such as the comma missing after the year.

Reviewer 3 Report

 The manuscript „Detection of Anaplasma phagocytophilum in wild and farmed  cervids in Poland“ deals with prevalence of A. phagocytophilum in wild and farmed Cervids from five sits. Samples of liver and spleen were analyzed using PCR that amplifies portion of around 380 bp of 16S rRNA.  The presence of A. phagocytophilum in wild animals from Poland was described in several paper so presence in cervids is not surprising. The highlight of current study is detection of potentially new genotype in roe deer identical to Spanish genotype. Unfortunately authors didn't use more genes to have stronger evidence of new genotype since amplified fragment is short. Furthermore authors didn't present differences between genotypes. Manuscript is clearly written but needs novelty and some parts should be improved.

Authors are using term Anaplasma spp. and A. phagocytophilum through the manuscript despite the fact the A. phagocytophilum was only Anaplasma detected in the current study.

Introduction is to general mostly focused on cervids rather than on A. phagocytophilum and part on clinical signs and course of disease should be re-written.  Same basic on genetic diversity is lacking.

M&M

Who was responsible for tissue collection and how it was performed? Authors didn't explain how the age of the animals was determined and  divided in groups of young and adults. What was the age limit for animals to be classified in the group of young? It is not clearly presented how many animals originated from farms and how many were from wild population. Season of collection could have large impact on prevalence but is lacking in current study.

Were PCR products sequence in both directions and have authors noticed double peaks in electrophegrams?

Results

Comparison of nucleotide differences is missing. It is not clear why authors added table with accession numbers!

Reviewer 4 Report

Dear editor and authors,

I carefully read the manuscript entitled “Detection ofAnaplasma phagocytophilumin wild and farmed cervids in Poland”. The study has a limited degree of novelty, however if the authors focus their manuscript on less studied/discussed aspects as I suggested below, the relevance and/or the interest for the readers may be increased.

General comments:

Abstract: Lines 14-16: I do not agree with this, there are several studies on A. phagocytophilum genetic variability which evaluate both the presence and prevalence of different genetic variants, including zoonotic strains (see the review of Matei et al. 2019). Accordingly, some cervidae, especially red deer seems to harbor zoonotic strains. Indeed, their potential as reservoir host (i.e. by evaluating all three main components: tick production, realized reservoir competence and host density) should be further studied.

The less studied aspects which may be highlighted in this study and properly discussed/explained are the following: (1) evaluation of two tissue samples (spleen vs liver) with the obtained moderate agreement indicating multiple sample testing for establishing the most closest to the “real” prevalence; (2) a lower expected prevalence in the farmed cervids (although expected, probably due to a lower exposer to ticks by the restricted areal, this difference should be deeper discussed, eventually including other studies results); (3) the higher prevalence obtained in younger animals and the possible factors involved. I suggest to the authors to focus their manuscript, especially the discussions on these aspects in order to increase the relevance of their study.

In addition, the authors should discuss the limitation of the study, especially the low genetic discriminatory power of 16S rRNA. One or two nucleotide difference compared with the reference strain 16S rRNA fragment sequence do not reflect the high genetic variability observed when ankA, groEL or msp4 are used.

Specific comments:

Line 41-45: If refers at Anaplasma genus the vectors or probable vectors are from many other species. I suggest to limit the introduction to A. phagocytophilum.

Line 71-72: how many red deer were wild and how many farmed; the prevalence difference was statistically significant? For proper discuss this difference some details regarding the farming condition should be added: the farm area/animal density and the ecological details like forest coverage etc., treatments-especially for ectoparasites, type of fancying (it allows the access of other wild animals like medium sized mammals); etc.

Line 77: what was the established age limit to differentiate young from adults; in cervids the morphological characteristics (horns, teets, size) allow a quit exact estimation of age.

Line 91-92: p=0.069 is not at borderline of statistical significance in my opinion.

Line 110-111: may be mentioned that these three sequences from roe deer seem to have only one nucleotide difference (a C instead of T in the 758 position of the referenced strain), compared with the majority of sequence obtained. I suggest to change in: Altogether these four sequences may belong to the newly describedA. phagocytophilum Roe deer strain [ref? of MN170723.1].

Line 112: similar with above in roe deer: this sequence (I understand that was only one-is not clear) presented two nucleotide difference…

Line 117: this table highlight the low discriminatory power of the 16S rRNA, it can be referenced in the discussion part, where the limitation of this approach should be mentioned.

Line 119: the discussion can be improved as suggested in the general comments.

Line 223-224: A conclusion on the difference in prevalence can be drown only for red deer and only if there is a statistical significance. If not, it can be formulated as a simple observation, “seems to be more frequent…”

Author Response

Dear Reviewer, 

Kind regards

A.W. Myczka

Round 2

Reviewer 3 Report

Thank you for answering questions but changes didn’t improve quality of the manuscript. This is manuscript of local importance, regularly written but without real novelty.   

Please check definition of contig.

Contig: Contiguous sequence of DNA created by assembling overlapping sequenced fragments of a chromosome. A group of clones representing overlapping regions of the genome. A contig is a chromosome map showing the locations of those regions of a chromosome where contiguous DNA segments overlap. Contig maps are important because they provide the ability to study a complete, and often large, segment of the genome by examining a series of overlapping clones which then provide an unbroken succession of information about that region.

Contig is not the term used in assembling of two sequences.

Authors described clinical signs of anaplasmosis in wild animals without mentioning that  a  wide  range of wildlife species can be infected with A. phagocytophilum, but the impact of these infections on wildlife health is  unclear (André MR. Diversity of Anaplasma and Ehrlichia/Neoehrlichia agents in terrestrial wild carnivores worldwide: implications for human and domes‑tic animal health and wildlife conservation. Front Vet Sci. 2018;5:29). By presenting only clinical signs in cases it can be concluded that anaplasmosis is sever disease.  Authors just reshape sentence but in general nothing is changed.

It is difficult to understand explaination to comment  „Same basic on genetic diversity is lacking“

„The introduction does not contain information about genetic diversity of A. phagocytophilum 16S rDNA because the determination and analysis of genetic diversity were not the main aim  of the study. Additionally to the genetic diversity of the 16S rDNA gene, a different fragment of this gene is used than was used in this study. That is why the Authors did not include it in the introduction“

Author Response

Dear Reviewer, 

Kind regards 

A.W. Myczka

Reviewer 4 Report

Dear editor and authors,

The authors addressed to my comments. However, at least a small discussion on the genetic variability should be added if mentioned in the introduction. In fact the authors presented their sequencing results but did't discuss at all about it. As I mentioned in the previous comments there are enough evidence of the different strains circulating among wild ruminants and these were reviewed quit recently.

Author Response

Dear Reviewer 4, 

Thank you very much for the fact that you all have done so much work and devoted a lot of attention while reviewing our manuscript.

We send to you an improved manuscript, which has significantly increased in value and presents precise and reliable information about the research that we conducted.

Kind regards

Anna W. Myczka (Correspondent Author)
